# Climate Change: Water Temperature and Invertebrate Propagation in Drinking-Water Distribution Systems, Effects, and Risk Assessment

**Günter Gunkel** [1,*] , **Ute Michels** [2] **and Michael Scheideler** [3]

1   Inwert Institute for Biological Drinking Water Quality, 13465 Berlin, Germany
2   Equality's, 15745 Wildau, Germany; utemichels@aqualytis.com
3   Inwert Institute for Biological Drinking Water Quality, 45721 Haltern, Germany; ms@scheideler.com
*   Correspondence: guenter.gunkel@water-quality-control.de

**Abstract:** This paper provides a summary of the knowledge of drinking-water temperature increases and present daily, seasonal, and yearly temperature data of drinking-water distribution systems (DWDS). The increasing water temperatures lead to challenges in DWDS management, and we must assume a promotion of invertebrates as pipe inhabitants. Macro-, meio-, and microinvertebrates were found in nearly all DWDS. Data in relation to diversity and abundance clearly point out a high probability of mass development, and invertebrate monitoring must be the focus of any DWDS management. The water temperature of DWDS is increasing due to climate change effects, and as a consequence, the growth and reproduction of invertebrates is increasing. The seasonal development of a chironomid (*Paratanytarus grimmii*) and longtime development of water lice (*Asellus aquaticus*) are given. Due to increased water temperatures, a third generation of water lice per year has been observed, which is one reason for the observed mass development. This leads to an impact on drinking-water quality and an increased health risk, as invertebrates can serve as a host or vehicle for potential harmful microbes. More research is needed especially on (i) water temperature monitoring in drinking-water distribution systems, (ii) invertebrate development, and (iii) health risks.

**Keywords:** biological stability; DWDS; health risk; water lice *Asellus aquaticus*; chironomid larvae *Paratanytarus grimmii*; *Thermocyclops oithonoides*

## 1. Introduction

Maintaining and safeguarding good drinking-water quality is a complex challenge, beginning with raw water protection (groundwater, surface water, and wells), raw water treatment in waterworks, and the management of drinking-water distribution systems (DWDS) comprising pipelines and storage facilities. In addition to technical faults, there are other risks, including contamination and eutrophication of surface waters, warming of lakes and rivers, and groundwater contamination. Disturbances in raw water treatment can occur, e.g., as a result of extreme water consumption peaks during extended dry summer periods.

The impact of temperature increase on drinking-water quality is not yet the focus of the climate change debate, while a comprehensive study about water temperature in DWDS points out that there is clearly insufficient knowledge and a lack of analysis [1]. In general, two modes of impact must be distinguished: first, influences on raw water quality and second, on temperature-dependent biological processes. Only some data are available on climate change effects on drinking-water quality—mainly on raw water availability and quality [2]—but analyses of temperature increase effects on water quality in DWDS are scarce.

In DWDS, the impairment of water quality is observed, in particular, due to the mass development of microbes and/or small invertebrates as pipe inhabitants. Some of the

more or less unaesthetic invertebrates are midge larvae, snails, water lice, and worms, such as oligochaetes and nematodes. Only a few studies have been carried out about DWDS water temperatures and microbial growth [3], the health risk of increasing water temperature [3,4], and invertebrate abundance in DWDS, e.g., in the Netherlands [5], Denmark [6], and Germany [7].

Temperature increases in ground water can exceed the temperature tolerance of groundwater invertebrates [8] and promote biochemical processes with increased oxygen consumption; thus, the risk of anoxic groundwater is increasing.

In soil, the effects of seasonal temperature changes occur down to 10–15 m depending on the groundwater level and geomorphology. Thus, in the long term, an increase in ground-water temperature can be expected. Measurements in Germany in a reference area confirm this and have already detected a significant increase in groundwater temperature in the deeper neutral zone (>40 m depth, free of seasonal temperature effects). At a 40 m depth, an increase of 1.7 °C from 1984–2015 was observed [9]. Some authors report comparable temperature increases of 0.28 °C per decade in 20 m depth and of 0.16 °C in 40 m depth [10], respectively, of 0.1–0.4 °C per decade [11]. Additionally, spring-water temperatures increased significantly by 0.5–1.5 °C per decade in Austria as a mean and in Germany/Hessen by 0.9 °C per decade [12].

Increasing water temperatures of rivers mainly in shallow littoral zones, where ground-water recharge occurs, can impact bank filtration efficiency because water temperature regulates bio-chemical processes. Key factors for bank filtration efficiency are the temperature tolerance of organisms, increasing biomass of interstitial flora and fauna, intensification of turnover of organic carbon, and, as a consequence, increased biochemical consumption of oxygen [13]. In general, we can apply the temperature rate of the reaction rule; this means that an increase of 10 °C leads to biochemical reactions two or three times faster, which corresponds to an increase of 10–20% per °C. In addition, climate change effects, such as long-lasting droughts or flood events, influence bank filtration, but only few studies are available [13,14]. Concerning direct water abstraction, these climate change effects impact water quality, too.

For many years, climate change effects on lakes and reservoirs have been the focus of limnological research, and there is no doubt that an increase in water temperatures in lakes and rivers has already occurred. The annual average water temperature increased in large European rivers by 1–3 °C over the last century. The average surface-water temperature increase of European lakes is in the range of 0.5 to 0.6 °C per decade [12]. This is particularly relevant for direct drinking-water abstraction from surface waters and in case of bank filtration and for groundwater recharge.

For a few decades, the high complexity of limnic systems with increasing water temperatures have been recognized as leading to substantial changes in physical, biochemical, and biological processes. The ongoing warming of water bodies in summer and missing ice covers in winter periods, with a lack of stratification in lakes and reservoirs, leads to profound changes of limnic ecosystems [12,15]. The increased stability of thermal stratification in summer leads to an extension of the duration of the stratification period. The epilimnion (upper water body) is impoverished in nutrients with the promotion of cyanobacteria; the hypolimnion (deep water body) becomes warmer, more nutrient-rich, and oxygen-poor. Organisms sensitive to these parameters can react, e.g., by diversity and abundance as well as by seasonal succession of species. In particular, the increased occurrence of cyanobacteria with increasing lake heating is a severe problem for water quality, as many cyanobacteria produce highly effective toxins.

In DWDS, the water temperature can significantly increase during transport from the water-work outlet to consumer. The DWDS temperature is strongly influenced by climate, especially urban heating, depth of pipe installation, soil surface (asphalt and vegetation), ground-water level, and operation parameters, such as water flow, stagnation periods, etc. [1].

Change of drinking-water quality at the transport from water work to consumer is described by biological stability. In 2006, the World Health Organization (WHO) pointed out the importance of biological stability of drinking water, particularly in the context of microbiological stability and safety [16,17]. A more comprehensive assessment of biological stability as a complex, multifactorial process is available [3]. It encompasses interactions between water temperature, water chemistry, dissolved and particulate organic substances, pipe material and structure, biofilm formation, and pipe colonization by invertebrates.

Invertebrate development and spreading in DWDS must become a greater focus of water-quality management because invertebrates can serve as a carrier or host for harmful microbes. Microorganisms can settle in the gut of invertebrates as a host and are even able to multiply and form a species-specific biome, well-known from chironomids, nematodes, and amoebae [18–20].

Bacterial and invertebrate communities are key factors for the biological stability of piped water, but until now, little has been known about population regulation and limiting factors. There are only a few studies available, e.g., development of biofilms in DWDS [21], interaction of microbes and invertebrates [22], invertebrate body size classes and biomass [23], and promotion of microbes by trace substances of invertebrate excretion [24].

The analyses of food limitation of pipe inhabitants point out that clearly, in general, the regrowth of biofilm exceeds daily food uptake of water lice. Only in the case of populations densities of several hundred animals per $m^3$ pipe volume can biofilm regrowth become a limiting factor [7]. However, other food resources must be considered, such as dissolved organic matter (DOM) and fine particulate organic matter (FPOM) from water works. Thus, water temperatures come more into focus.

Drinking-water pipes present a habitat with more or less long-term stable conditions concerning food availability, water quality, flow regime, and habitat structure (Figure 1), while only water temperature possesses a seasonal variance.

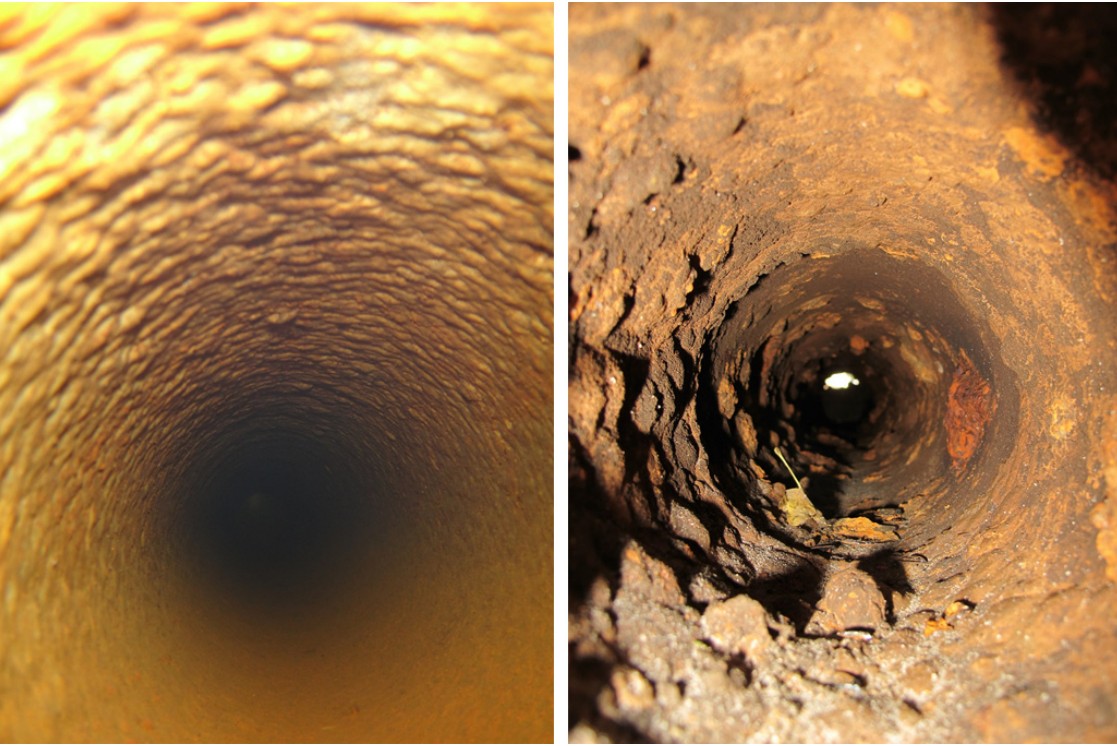

**Figure 1.** View inside drinking-water pipes; **left** side with small-scale incrustations and biofilm layer as a habitat for invertebrates (ND = 100); **right** side with corrosion of a cast iron pipe (ND = 80).

Some typical views inside drinking-water pipes are given in Figure 1, representing a small-scale structured habitat akin to a hilly landscape, and the inner wall is covered

with corrosion products (mainly in cast iron pipes), precipitations (carbonates, iron and manganese), and biofilm.

Thus, in DWDS, the development of invertebrates is regulated by water temperature, mainly via natural mortality, the growth rate, the beginning of reproduction (this means the number of generations per year), and fertility.

This presented research analyzes the development of invertebrates in drinking-water networks, and points out the increased water temperatures in DWDS. The data clearly highlight that invertebrate populations increase with increasing water temperatures. Overall, an impairment of water quality in DWDS due to promotion of invertebrates must be discussed. Concerning water lice, the increase in population growth is caused by the development of a third generation per year.

## 2. Methods

### 2.1. Sampling of Invertebrates

For about 20 years, DWDS have been analyzed for water-quality control based on hydrant sampling, predominantly located in Germany and the Netherlands, including groundwater, bank filtration, and high mountain reservoirs as a source of raw water. Chlorination of drinking water in the studies DWDS has not been performed. The database comprises more than 2000 hydrant samples. Samples were collected from DWDS with known occurrence of invertebrates and without any abnormalities as well as from several years' monitoring of a DWDS in North Germany. In total, 157 different hydrants were sampled in 27 different DWDS (24 in Germany and 3 in the Netherlands).

Sampled pipes were of ND 60–315 mm, but most had a dimension of ND 80–150, and thus, a 1 m$^3$ hydrant sample corresponds to 200–60 m pipe length. Different pipe types were sampled (PE, HP-PE, PVC, asbestos cement, cast iron), but up to now, no effect of pipe material on invertebrate settlement was observed. Some other pipe parameters, such as pipe age and accumulation of detritus (sedimentation volume), were partly available, but no effect on invertebrate development was recognizable. Flow rate is of importance for the development of invertebrates with species-specific tolerance ranges. However, the flow rate in DWDS is a high dynamic parameter with day/night cycle and stagnant zones, influenced by periodic or singular events, such as pipe flushing, firefighting, hydrant use, etc. Thus, no resilient data of flow rates are available. A more detailed classification of the sampled pipes could not be conducted because many of the related parameters, such as intensity of pipe maintenance, position of the hydrants (on side/on top), incrustations and intensity of biofilms, variations in flow direction, and high flow events caused by firefighting, were not available.

DWDS were analyzed using a standardized sampling procedure for hydrants with a volume of 1 m$^3$ and a flushing rate 1 m s$^{-1}$; pipe flow was unidirectional and provided by separated pipe sections. This corresponds to the rules of the German Technical Paper W 271 [25].

A low-pressure high-flow through stainless-steel filter was developed to separate invertebrates without causing damage [26]. A mesh size of 100 μm was used for macro- and meioinvertebrates and of 25 μm for the detection of microinvertebrates. The capacity of the filter was >30 m$^3$ h$^{-1}$, which enabled a high flow rate during sampling. The final filtrate volume was reduced to less than 300 mL and preserved with ethanol to a final concentration of 70% for microscopic analyses.

Several monitoring programs were conducted to study short-term and long-term development of invertebrates in DWDS; the locations had to be anonymous; thus, they are named A to E; main data of the studies DWDS are summarized in Table 1. Since 2013, long-term yearly monitoring of hydrants was carried out to analyze macro- and meioinvertebrates abundance (DWDS A). The focus of this study was the spreading and development of water lice and oligochaetes in unflushed and air–water flushed network sections. Since 2020, monthly monitoring of 6 hydrants in a DWDS has been performed to analyze chironomid larvae abundance and reproduction (DWDS B). Since 2001, weekly temperature registration in a particle filter was completed in DWDS C. In 2014, temperature

variance in a DWDS was studied, while hydrant sampling was performed (DWDS D). Temperature registration with two data loggers installed at the pipe wall and soil surface commenced in May 2020; in DWDS B, the pipe was constructed at a depth of 1.15 m. DWDS E was studied in 2016/2017 to analyze occurrence of copepods in tap water.

**Table 1.** Main water parameters as mean, respectively, and minimum/maximum of the studies DWDS (data of the water associations).

| Parameter | DWDS A | DWDS B | DWDS C | DWDS D | DWDS E |
|---|---|---|---|---|---|
| Raw water source | Groundwater | Groundwater | Groundwater | Groundwater | High mountain reservoir |
| Drinking-water treatment | Aeration, filtration | Aeration, flocculation, filtration | Filtration, aeration | Aeration, filtration | Filtration |
| Chlorination | None | None | None | None | Yes |
| Temperature (°C) | | 9.9–11.1 | 11.1–14.7 | 8.0–20.7 | 5.0–11.1 |
| Conductivity ($\mu S\,cm^{-1}$, 25 °C) | 455–659 | 255 | 784 | 678 | 16 |
| pH | 7.4–7.7 | 7.9–8.1 | 7.2 | 7.4 | 8.9–9.2 |
| Turbidity (NTU) | <0.15 | 0.2 | | | |
| Colour (SAC 436 nm) | 0.2 | 0.48 | | | 0.05 |
| Fe total (mg $L^{-1}$) | <0.01 | 0.08 | 0.01 | 0.01 | <0.001 |
| TOC | 1.4–2.4 | 6.1 | 4.3 | 2.4 | 1.6 |

### 2.2. Invertebrate Analyses

Macroinvertebrates are classified as >2 mm in size and distinguishable with the naked eye, such as water lice, oligochaetes, snails, and chironomid larvae [27]. They were determined under optical magnification (Olympus SZX 16 and Olympus SZ 40), measured and counted. Systematic analysis was generally conducted to species level (but not for small oligochaetes and springtails). For water lice, the size, sex, molting stages of females, and egg and embryo number were recorded. Thus, abundance, biomass, growth, and reproduction were calculated. The results were used to build up a databank with about 2000 data sets to evaluate invertebrate abundance.

Meioinvertebrates comprise of small animals 0.1 mm to 2 mm in size, including water flees (Cladocera), copepods, and nematodes. Microinvertebrates are small animals with a size of 0.025 mm to 0.1 mm and are represented by rotifers, ciliates, and amoebas. Both size classes were analyzed using a transmitted light microscope (Olympus BX 40, Olympus, Hamburg, Germany) and an inverse microscope (Olympus IMT-2 and CKX 41, Olympus, Hamburg, Germany); abundance and biomass were also calculated. Systematic analyses were performed to species level only for the copepods and phyllopods.

The number of invertebrates were calculated as empirical *p*-quantile statistics with 10th percentile (=10% of the samples), median (=50% of the samples), and 90th percentile (=90% of the samples). A significance analysis of co-relationships was performed by using the Pearson function.

### 3. Results

*3.1. Temperature Increase in Drinking-Water Reservoirs*

In reservoirs, the plankton community is changing due to the water-temperature increase: first by dominance of cyanobacteria and second by modified life cycles of some zooplankton species and the distribution of these species in the water column of reservoirs. Cyanobacteria show a clear tendency to dominate lake plankton at higher water temperatures; in addition to increased irradiation, prolonged thermal stratification in lakes and reservoirs is also discussed as a key factor [15,28,29]; in addition, odor and taste substances excreted by cyanobacteria are also relevant to drinking water, such as Geosmin and MIB (methylisoborneol) [30–34].

A change of life cycle with severe effects on drinking-water quality has been observed for *Thermocyclops oithonoides*, a copepod, in a drinking-water reservoir in northern Germany, with water abstraction without micro sieving. *Thermocyclops oithonoides* (adults of which are 0.7 to 1.0 mm in size, and juvenile stage, so-called copepodite, are in 0.1–0.7 mm in size) is a typical representative of the open water of lakes and exhibits a distinct life cycle in the upper water body, the epilimnion (from approximately April to October), and a so-called diapause during overwintering in the lake sediment (November to March/April). With this life cycle, the pelagial (deeper water body) and the water abstraction layer are free of *Thermocyclops oithonoides* all throughout the year.

For a number of years, however, overwintering has been observed in the pelagial of lakes as so-called pelagic diapause, presumably triggered by higher water temperatures in recent years without ice formation [35–37]. Thus, in winter, raw water was enriched with this copepod, and the overwintering C5 copepodite stage was introduced in large numbers into DWDS. In 2016, the first records of the pelagic diapause of *Thermocyclops oithonoides* occurred in a larger drinking-water reservoir in northern Germany and led to a severe impact on drinking-water quality and tap-water quality; up to 2000 *Thermocyclops oithonoides* per m$^3$ were registered in the DWDS. Only a microfiltration with a mesh size <0.025 mm can inhibit this contamination.

### 3.2. Temperature Increase in DWDS

### 3.2.1. Daily Variation of Water Temperatures in DWDS

Beside long-term increasing water temperatures in DWDS caused by raw water and/or heating of the soil, additional short-term effects of temperature increases have occurred. Heating of the soil leads to a daily oscillation of drinking-water temperature in DWDS B, located in northern Germany, where a datalogger was installed (Figure 2). Variation of surface soil temperature was in the range of 14 to 21.8 °C and led to daily oscillations of the drinking-water temperature of up to 1.7 °C. The day/night temperature variance is visible in the pipe's water temperature; thus, soil surface heating is the triggering factor. Hot soils directly lead to drinking-water heating with a lag time of a few hours; e.g., 12.1 mm of rainfall on 24 July led to a decrease of soil surface temperatures and more or less constant water temperatures in the pipe.

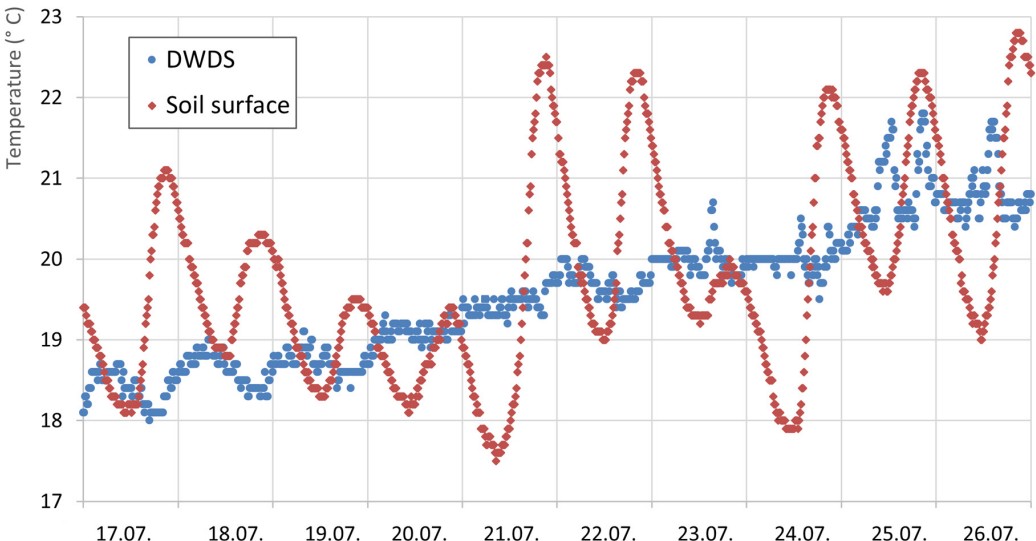

**Figure 2.** Daily temperature variance in DWDS B in Germany/Lower Saxony; pipe is 1.15 m in depth; the soil surface is stony/sandy.

Several parameters are known to regulate the water-heating processes, such as the soil surface (asphalt, stones, sand, and vegetation), soil moisture, ground-water level, and flow rates in the drinking-water pipe, but these data were not available.

### 3.2.2. Seasonal Temperature Increase

The heating of water in DWDS is triggered by seasonal effects, and increased soil surface temperatures lead to increased temperatures in drinking-water pipes (Figure 3). Temperature increases in a DWDS can be high, with a monthly increase of about 3 °C in springtime. Thus, climate change effects with heat spots in urban areas and heating of soil surfaces lead to an increase of a pipe's water temperatures, and available guidance values of 20 °C to 25 °C can be exceeded.

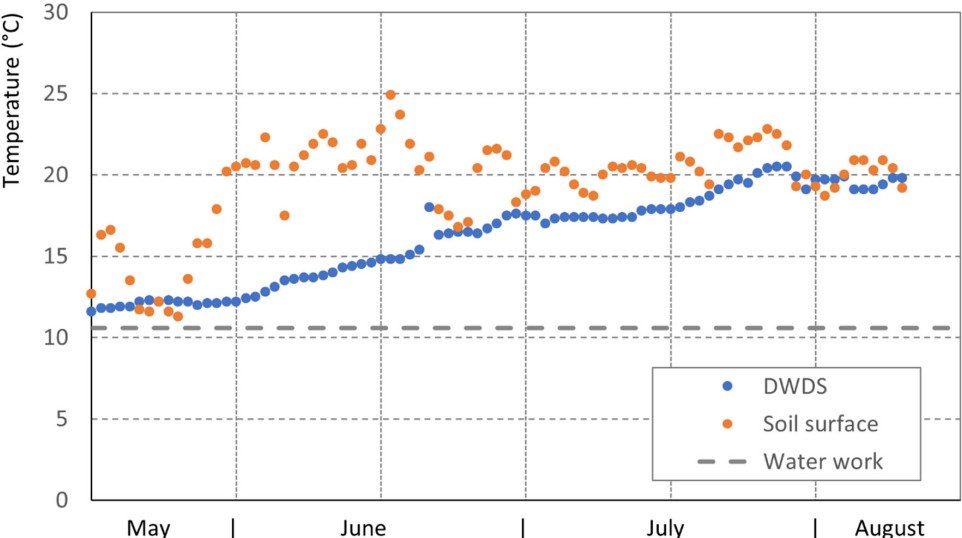

**Figure 3.** Seasonal heating of drinking water in Germany/Lower Saxony (DWDS B); water works outlet temperature, the temperature of the soil surface, and the temperature increase of the drinking-water pipe are all given; pipe DN in 1.15 m depth.

### 3.2.3. Long-Term Temperature Increase in DWDS

The water temperatures in DWDS are determined not only by the raw water temperature but also by the heating of the upper soil layer in which drinking-water pipes are laid. Data about water temperatures in DWDS are hardly available; water temperatures are usually only recorded at the outlet of the waterworks. One long-term data series was taken from Germany/Brandenburg (Figure 4) and clearly shows the seasonal oscillation and a rising trend of temperature with a warming of 1.5 °C in 20 years and a temperature increase from 11.0 °C (2001) to 12.5 °C (2021). High water temperatures of about 20 °C sometimes occur, but only weekly data were available; thus, a more detailed analysis of temperature maxima was not possible.

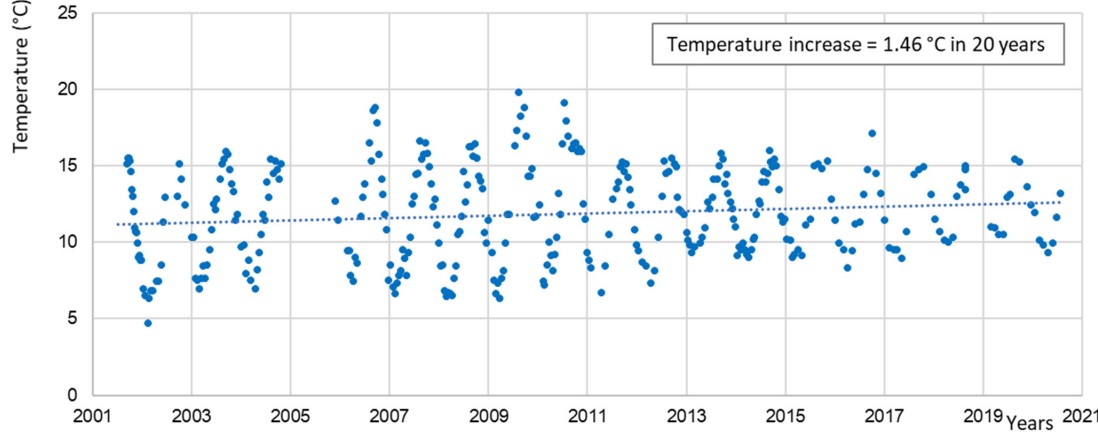

**Figure 4.** Long-term water temperature increase in DWDS C in northern Germany/Brandenburg; registration was performed weekly in a particle filter in the DWDS, mod. from [37].

### 3.2.4. Temperature Variance in a DWDS

Water temperatures in a DWDS in northern Germany/Lower Saxony varied with the distance from the waterwork, leading to an increase in the summer period from 12.6 °C in the waterwork to 20.8 °C in the DWDS, 2.3 km from the waterwork; this means a heating by 8.2 °C during water distribution (Figure 5). The registered temperatures in the DWDS clearly point out several areas with increased water temperatures; besides the distance to the waterwork, other parameters, such as the flow rate (pipe dimension and water consumption), seem to be key factors for water heating. Increasing water residence times support the water-heating process, but this is a multi-parameter factor (intensity and day/night cycle of water consumption, pipe dimension, flow direction changes in the pipe network); only complex modeling of the drinking-water network can give us data but were not available here.

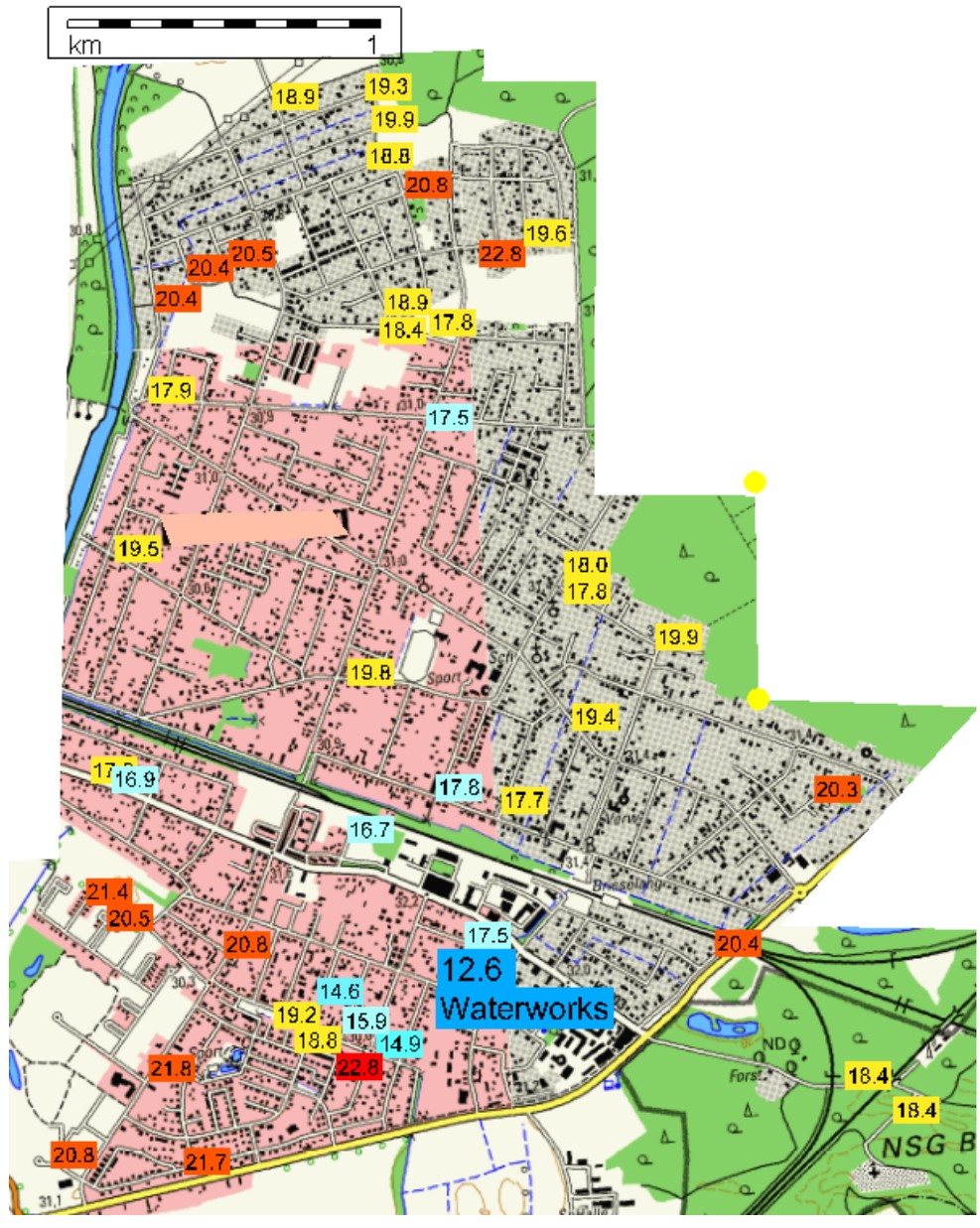

**Figure 5.** Water temperature variance in DWDS D in northern Germany/Lower Saxony in September (15.9–25.9.2014), dark blue = waterworks with 12.6 °C; light blue = 15.0–17.5 °C; yellow = 17.6–20.0 °C; red ≥ 20 °C.

### 3.3. Diversity and Abundance of Invertebrates in DWDS

Invertebrates are found in all DWDS; small invertebrates, meioinvertebrates (0.1–2 mm), and microinvertebrates (0.025–0.1 mm) are mainly found, originating from filters of the waterworks. However, many DWDS are also settled with macroinvertebrates (>2 mm), which are characteristic for surface water and do not originate from wash out of the filters. Diversity and abundance of invertebrates living in drinking-water networks is high (Table 2).

**Table 2.** Size range and occurrence of invertebrates in DWDS in the European lowlands. Data base: macroinvertebrates = 1039 hydrant samples; meioinvertebrates = 1320 hydrant samples; microinvertebrates = 584 hydrant samples (modified and actualized from [7]). Ind-individuals.

| Animal Groups and Species | Size Range (Length) (mm) | Occurrence Probability (%) | Median (ind. m$^{-3}$) | 90th Percentile (ind. m$^{-3}$) | Maximum (ind. m$^{-3}$) |
|---|---|---|---|---|---|
| Macroinvertebrates (total) | >2 | 96.5 | 15.9 | 135 | 4764 |
| *Isopods* Water louse (*Asellus aquaticus*) (Proasellus coxalis) | 0.5–11 | 79.3 | 15.6 | 61 | 869 |
| Cave water louse (*Proasellus cavaticus*) | 1–6 | sporadic | 8.0 | 34 | 89 |
| *Amphipoda* Freshwater amphipod (*Niphargus aquilex*) | 0.4–6.5 | rare | 1.9 | 14.6 | 40 |
| Midges, adults (*Simuliidae*) | 1–4 | sporadic | | | |
| Chironomide, larvae (*Paratanytarsus grimmii*) | 5 | rare | 27 | 154 | 1834 |
| Chironomide, adults, larvae (*Limnophyes asquamatus*) | 2 | sporadic | | | |
| *Oligochaete* earthworms (*Oligochaeta*) | 0.5–40 | 74.9 | 6.0 | 92.3 | 4723 |
| Springtails (*Collembola*) | 1–5 | sporadic | | | |
| *Snails* Nautilus ramshorn (*Gyraulus crista*) | 1–2 | rare | 6 | | 1599 |
| Bladder snail (*Physella acuta*) | 1.5–5 | rare | | | |
| New Zealand mud snail (*Potamopyrgus antipodarum*) | 1–3 | rare | | | |
| *Bryozoa* (*Plumatella* spec.) | 20 | sporadic | | | |
| Meiofauna (total) | 0.1–2 | 98 | 1703 | 14,522 | 336,010 |
| *Copepods* | | | | | |
| *Juveniles and adults* | 0.1–1.4 | 82 | 422 | 3923 | 157,643 |
| *Nauplien* | 0.1–0.5 | 72 | 360 | 3107 | 23,467 |
| Roundworms *Nematoda* | 0.1–4 | 61 | 1058 | 9294 | 336,000 |
| Water mites *Hydracarina* | 0.1–0.8 | 60 | 117 | 1097 | 14,331 |

Table 2. *Cont.*

| Animal Groups and Species | Size Range (Length) (mm) | Occurrence Probability (%) | Median (ind. $m^{-3}$) | 90th Percentile (ind. $m^{-3}$) | Maximum (ind. $m^{-3}$) |
|---|---|---|---|---|---|
| Water fleas (*Cladocera*) | 0.1–0.8 | 54 | 333 | 5149 | 68,400 |
| Water bears *Tardigrada* | 0.1–0.5 | 15 | 65 | 446 | 4267 |
| Hairybellies (*Gastrotricha*) | 0.1–0.2 | 14 | 273 | 2.011 | 15,600 |
| Flatworms (*Turbellaria*) | 0.14–1.4 | 12 | 68 | 404 | 3252 |
| Seed shrimps (*Ostracoda*) | 0.15–0.8 | 8 | 77 | 807 | 25,445 |
| Microfauna (total) | 25–100 | 100 | 75,381 | 465,589 | 10,426,389 |
| Rhizopods (*Testaceae*) | 0.02–0.16 | 100 | 72,339 | 456,980 | 9,976,389 |
| *Rotatoria* | 0.05–0.4 | 89 | 3096 | 15,314 | 472,000 |
| *Ciliata* | 0.05–0.1 | 6 | 215 | 1020 | 6667 |
| Naked amoebae (Free living amoebae, FLA) *Amoebina* | 0.05–0.2 | 3.4 | 354 | 4058 | 4636 |

Macroinvertebrates were found in 96.5% of the hydrant samples with a median abundance of 15.9 ind. $m^{-3}$ water; the 90th percentile amounts to 135 ind. $m^{-3}$, and the maximum is 4764 ind. $m^{-3}$. The data clearly point out that many of the hydrant samples (28%) exceed quality limits of 40 ind. $m^{-3}$. Some of the macroinvertebrates originate from groundwater, e.g., the isopod *Proasellus cavaticus*; others are typical inhabitants of surface water, e.g., the isopods *Asellus aquaticus* and *Proasellus coxalis* and snails (*Gyraulus crista*, *Physella acuta,* and *Potamopyrgus antipodarum).* Midges can enter DWDS via egg deposition of flying adults. However, it must be pointed out that in general, only a few animals enter the DWDS, and the high abundance can only occur if these invertebrates are able to survive and to reproduce in the DWDS.

Meio- and microinvertebrates were found in all hydrant samples with high abundance; the median of meioinvertebrates is 1703 ind. $m^{-3}$ (90th percentile 14,522 ind. $m^{-3}$), and the microfauna is much more present, with a median of 75,381 ind. $m^{-3}$ and a 90th percentile of 465,589 ind. $m^{-3}$.

*Proasellus coxalis* is an isopod, originating from the Mediterranean region, which migrated into Northern European countries in recent decades (Figure 6). In 2020, *Proasellus coxalis* was observed in a DWDS in northern Germany, which was settled in 2019 only by *Asellus aquaticus*. In 2020, the portion of *Proasellus coxalis* amounted to about 50% of the adult animals (the juveniles cannot be distinguished by species) and reached a high abundance with 21 ind. $m^{-3}$ (maximum 146 ind. $m^{-3}$). In 2021, *Proasellus coxalis* reached 90% of abundance with a maximum of 173 ind. $m^{-3}$.

The New Zealand mud snail *Potamopyrgus antipodarum* is another invasive species found in a German DWDS. *Potamopyrgus antipodarum* originates from New Zealand and is spreading in European countries (Figure 6). The snail reaches up to 3 mm and reproduction occurs by parthenogenesis, and thus, a rapid spreading must be assumed.

Climate change and global trading leads to migration in surface waters, namely the transport of invertebrates into new, not yet populated areas; in general, these invasive species impact natural fauna. It is of interest that some invasive species can settle DWDS, too, in general by contamination via surface water and in some cases directly by contamination from back siphonage of water by refilling ship tanks, fire, and watering trucks.

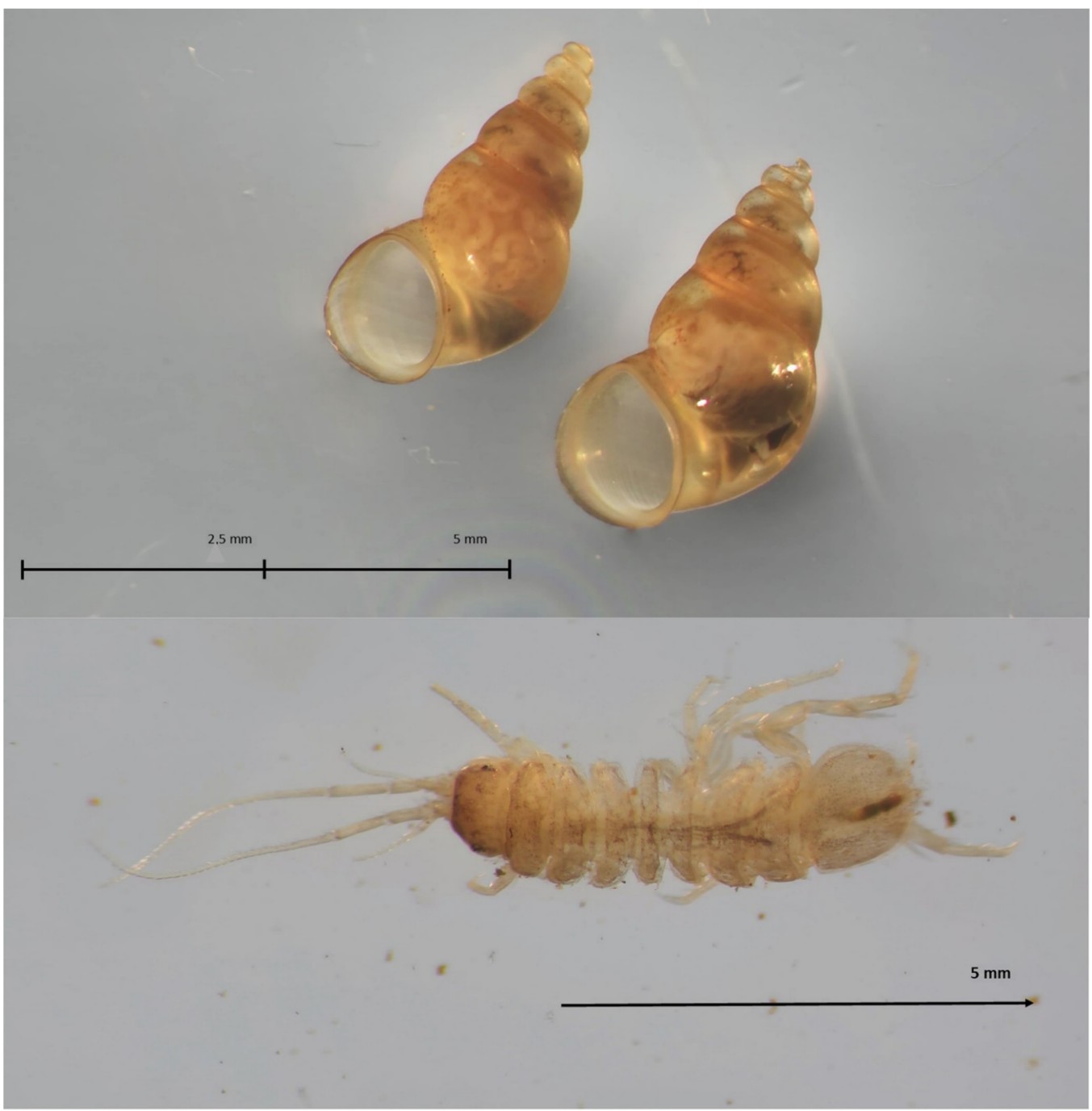

**Figure 6.** Invasive species in German DWDS: (**above**) New Zealand mud snail *Potamopyrgus antipodarum* and (**below**) water lice *Proasellus coxalis*.

The very high abundance of pipe inhabitants (Table 2), e.g., 135 macroinvertebrates per m$^3$ as 90th percentile in DWDS fed with good water quality (mainly with ground water except for some systems with bank filtration and only a few systems with high mountain reservoirs), and a drinking-water treatment with the recognized state of the art is surprising. Even the maximum values are no outliers but instead are part of a normal distribution [7].

Key factors for the high abundance of invertebrates in DWDS [7] are the conditions in the pipe as a habitat with physical–chemical and biological parameters. Biological regulation by predators is restricted to some meioinvertebrates, e.g., some copepods and water mites.

### 3.4. Seasonal Development of the Chironomid Larvae Paratanytarsus grimmii in a DWDS

A yearly cycling of abundance is observed for *Paratanytarsus grimmii*, which is a chironomid (Figure 7). Temperature is the only parameter in the DWDS underlying seasonal variation, while light, available food, etc., are more or less constant parameters. Thus, abundance cycling of invertebrates in DWDS must be triggered by water temperature, similar to seasonal variation in surface waters. *Paratanytarsus grimmii's* growth, reproduction, and fertility is regulated by temperature, and high abundance is observed in August and November when temperatures increase. In the winter period, with temperatures <12 °C, overwintering occurs with increased mortality. The seasonal variation of *Paratanytarsus grimmii* is very high with means of 48 ind. m$^{-3}$ in the winter period and 235 ind. m$^{-3}$ in the summer period.

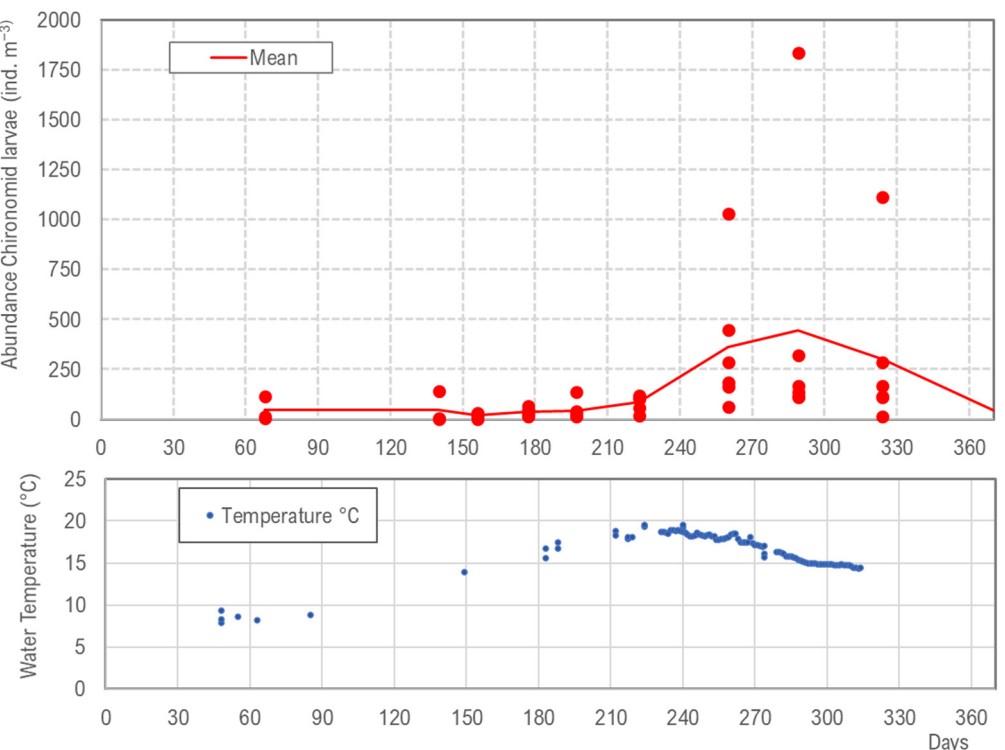

**Figure 7.** (**Above**) Abundance of chironomid larvae (*Paratanytarsus grimmii*) in DWDS B (Lower Saxony, Germany) and data of six hydrants monitoring in 2020; (**below**) water temperature in the DWDS and data of water samples and data logger, mod. from [37].

The high increase of *Paratanytarus grimmii* abundance is due to the high number of generations per year; in northern Germany, five generations were recognized by size distribution of the larvae. In spring and early summer, two generations were observed (May and June), but the population increase was only moderate. In late summer and autumn, with higher water temperatures, three generations were registered (August, October, and November), which led to the significant increase of the abundance.

### 3.5. Long-Term Development of Water Lice

There are currently limited data available on the long-term development of invertebrates in DWDS, as multi-year monitoring programs have been carried out only rarely. One exception was hydrant monitoring by a water company in northern Germany from 2013 to 2019. The occurrence of water lice (*Asellus aquaticus*) shows a continuous increase in density; the median population densities increased from 0 ind. m$^{-3}$ to 13 ind. m$^{-3}$, and the 75 ind. m$^{-3}$ percentile increased from 3 to 50 ind. m$^{-3}$ (Figure 8). Moreover, a spatial extension is given from year to year; in 2013, 42% of the studies hydrants were settled with

water lice, while this portion had increased to >90% in 2019 (Figure 8). Drinking-water treatment and network operation were not changed during the study period, so it must be assumed that other parameters, such as water temperature, are of significance. Data of water temperature are not available, but locally, the annual mean air temperature increased from 8.6 °C in 2010 to 10.3 °C in 2019 ($r^2$ = 0.52) although the time period is too short for a statistical trend analysis.

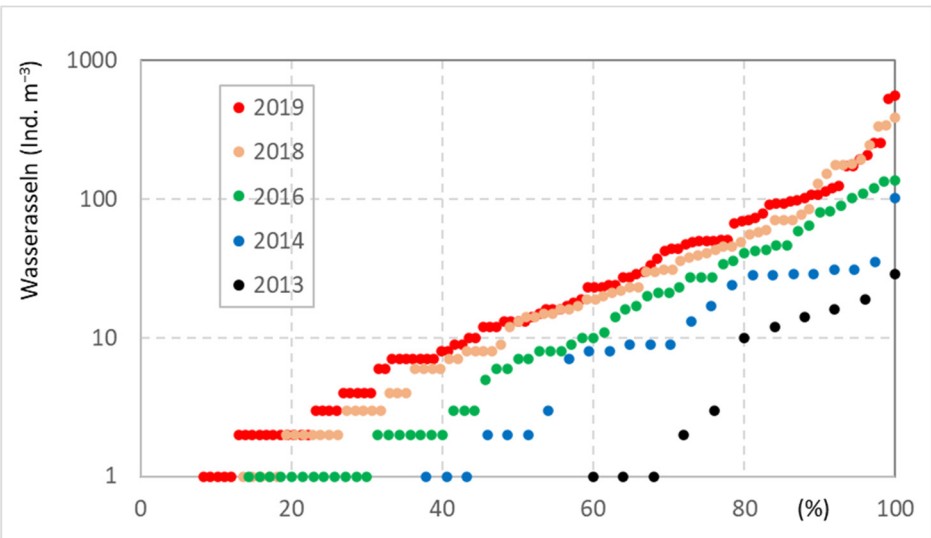

**Figure 8.** Increase of abundance of water lice (*Asellus aquaticus*) in a DWDS from 2013 to 2019, located in northern Germany/Schleswig Holstein; the distribution curves of the hydrant monitoring (DWDS A) in percentage (100% = total number of hydrant samples of one year) are given, mod. from [37].

The increasing trend of population density of water lice is given in Figure 9, presenting a marginal sub-region of DWDS A of about 15 km². Since the last pipe flushing with air–water in 2012, a linear lice population increase occurred with a lag phase of 1 year (y = 0.37x − 5.00; $R^2$ = 8.88), and up to now, no steady-state conditions are recognized, with 70 water lice m$^{-3}$. This population increase is given by an increased number of animals in the hot spots and an spatial extension within the sub-DWDS.

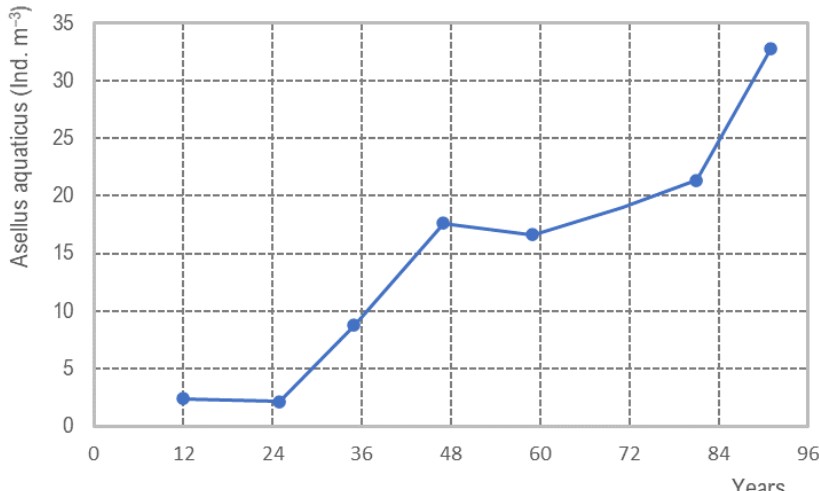

**Figure 9.** Increase of abundance of water lice (*Asellus aquaticus*) in a sub DWDS from 2012 to 2019, located in northern Germany/Schleswig Holstein; data are based on yearly nine hydrant samples of about 15 km² large sub DWDS; in 2012, air–water flushing was done, following years without any flushing activities.

A dynamic population analysis indicates clear growth and reproduction of *Asellus aquaticus* in February/March, July, and November (Figure 10). The parent generation in spring (the winter form) with large animals reproduced in March, and these water lice have about 40 embryos per female. Brood care of females leads to high reproduction success. A second generation is observed in July with smaller animals, and a third generation occurs at the end of the year in November. The number of eggs, specifically embryos, decreases to approximately 22 embryos per female in the summer and 15 embryos per female in the winter.

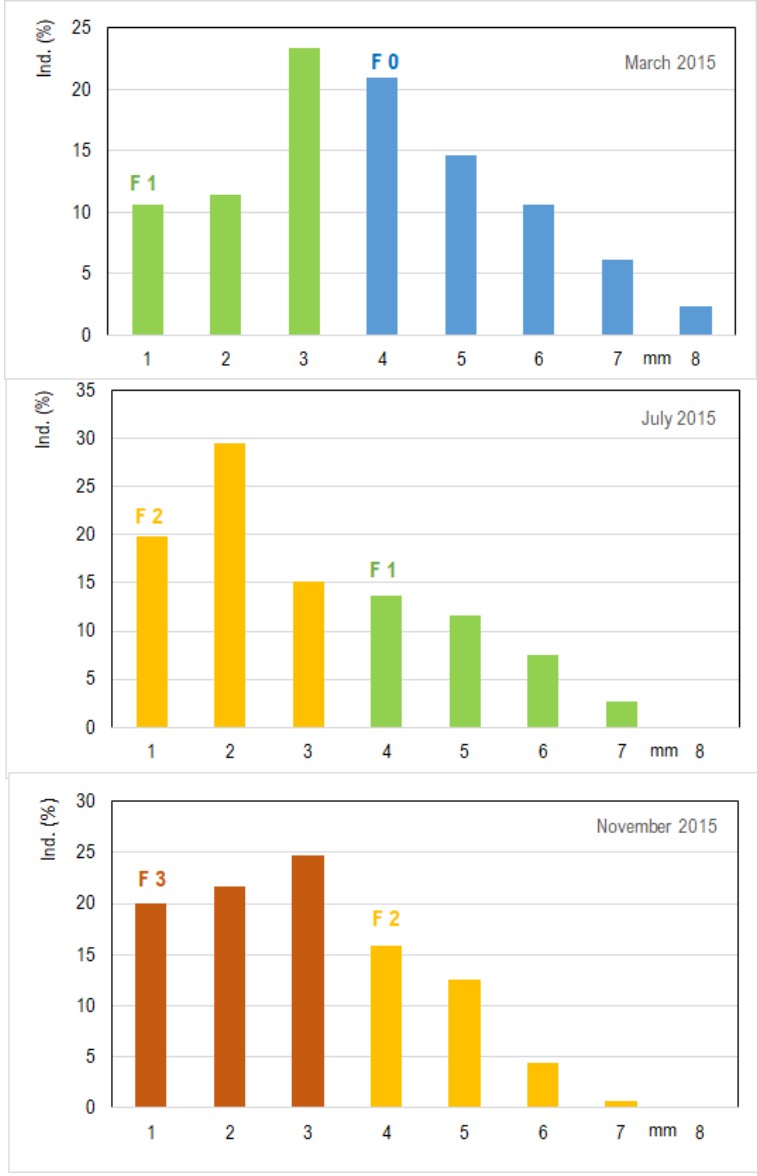

**Figure 10.** Development of water lice population in a drinking-water network in northern Germany/Schleswig Holstein in 2015; the proportion of the individual size classes in percent of the total population is given; F0 is the parent (winter) generation, and F1 to F3 denotes generations 1 to 3 (DWDS A). Coloring represents the different water lice generations: blue = F0 generation (0 overwintering animals), green = F1 generation, yellow = F2 generation, and brown = F3 generation.

Thus, population growth (with a relation of males to females of 1:1 and without mortality) can be calculated for three generation cycles: two water lice (winter generation) produce 40 water lice as the first generation, the second generation amounts to up to 440 animals, and the third generation increases up to 3300 water lice, which is an exponential growth.

This estimation of the fertility points out clearly the significance of a third generation for population growth and extension of settled area, shown in Figure 8.

## 4. Discussion

### 4.1. Climate Change Effects on Drinking-Water Sources

Several studies on the heating of lake bodies exist and clearly point out a temperature increase as well as the prolongation of thermal stratification periods in lakes and reservoirs. The temperature of epilimnic lake water bodies has already risen by about 2–3 °C in the last few decades, and a further increase of 2 °C must be assumed by 2050 [12]. The increasing temperature of the epilimnion of lakes and reservoirs affects raw water quality; the spring and summer heating of lakes and reservoirs and the occurrence of several weeks of heat waves with extreme temperatures in the epilimnion must be distinguished. The average intensity of lake heat waves worldwide, defined as relative to the period from 1970 to 1999, increased from 3.7 to 5.4 °C, but their average duration increased dramatically from 7.7 to 95.5 days [38].

An effect of increasing water temperatures is given by disturbances of bank filtration and groundwater recharge, which are both mainly supported by infiltration of the upper (littoral) zone. Increasing water temperatures lead to decreased oxygen concentrations with increasing oxygen consumption by organisms; this means the oxygen balance is worsened, and the risk of anaerobic conditions is raised. Studies on temperature effect on bank filtration were conducted in Lake Tegel, Berlin/Germany using heated enclosures, resulting in a water temperature increase of 5 °C and an increase of sediment temperature of 2–3.5 °C [13]. The redox state of the infiltration zone was changed and characterized by a decline of oxygen concentrations within the interstices. However, the sulfate reduction with the release of the toxic sulfide ion has not been observed as a worst case.

However, raw water abstraction from lakes and reservoirs occurs in general in the hypolimnion, while water heating occurs mainly in the epilimnic zone. Thus, the risk of increased raw water temperature is mainly triggered by the higher temperatures in the winter period without ice formation. The data of the hypolimnic temperature increases in Germany are in the range of 3 °C by the end of the century [39], which is less than the heating of the epilimnion.

### 4.2. Water Temperature

The temperature of drinking water is an important quality parameter; drinking water should be fit for human consumption and thus also sufficiently cold. The WHO [16,17] emphasizes the importance of cold drinking water and points to the promotion of bacterial growth and taste problems, without even specifying guide values for this. Temperature limits are varied; the German drinking-water rule is 25 °C [40], but others recommend water temperatures of only 20 °C to minimize the growth of potentially human-pathogenic microbes, such as *Legionella* spp. [41]. The Robert Koch Institute in Germany also specifies a temperature of 20 °C for this purpose [42]. Impairment of the taste of drinking water can already occur at >15 °C [43,44].

In European countries, drinking-water temperatures typically range between 3 and 25 °C, fluctuating strongly with seasonal effects and with the distance from drinking-water treatment plants. Thus, a great variance of drinking-water temperatures in DWDS can be expected, and detailed monitoring is necessary.

Temperature tolerance of groundwater fauna has been studied; groundwater organisms are composed of a high proportion of so-called cold-stenothermic organisms—animals that react very sensitively to temperature changes [8]. However, representatives of groundwater fauna are found only sporadically in drinking-water systems, predominantly where spring water is used. Nevertheless, the impact on ground-water fauna due to increasing temperatures must be evaluated critically because decreased biodiversity and an abundance of groundwater fauna will reduce the metabolization of dissolved organic matter

(DOM), which means increasing DOC concentrations in ground water. The bioavailable part of DOC is the main food source of pipe inhabitants.

During drinking-water treatment, a species-rich (and more temperature-tolerant) filter biocoenosis occurs, predominantly meio- and microinvertebrates; in general, rising temperatures can be more or less compensated. In the temperature range of 15–20 °C, no adverse or lethal effects on the filter biocoenosis are expected.

### 4.3. Growth of Pipe Inhabitants

Temperature is a key factor for growth and composition of DWDS microbes and invertebrate communities. Elevated water temperatures are often associated with increased bacteria abundance, such as coliforms and *Aeromonas* spp. [3]. It is reported five times more in the colony forming unit (CFU) of *Aeromonas* at temperatures above 18 °C than at lower water temperatures [4]. A characteristic is the development of high cell numbers in summer/early autumn, when temperatures exceed 20 °C. The temperature optimum of *Aeromonas hydrophila*, for instance, is about 30 °C [45], while *E. coli* has an optimum temperature range of 12–30 °C [3].

In relation to microinvertebrates, temperature effect is species-related. Studies on environmental parameters for free-floating amoebae (FLA) in drinking-water networks report a genus-specific temperature effect: *Vermamoeba* was correlated with temperature, while *Echinamoeba* and *Cryptodiffugia* were inversely correlated with temperature [19].

The temperature effect of the most abundant meioinvertebrates is controlled by the life cycle. Life span of copepods is long-lasting, at 3–6 months, and with a resting period in winter; thus, the temperature effect on reproduction is small (e.g., *Megacyclops* sp., *Eucyclops* sp.), and only one or two generations occur within one year. Chydorids (Cladocera) have a short life cycle of 2–3 months (e.g., *Alona* spp.) or only 2 weeks in small-sized species (e.g., *Monospilus dispar)*. Additionally, Cladocera have a high reproduction by parthenogenesis [46]. Thus, temperature increases in DWDS can lead to an increase of generation number and support mass development of these pipe inhabitants.

In relation to the chironomid *Paratanytarsus grimmii,* the temperature effect on growth and reproduction has been intensively studied. Growth starts at 8 °C, and the fastest growth is at 25 °C; a lethal temperature is as high as 30 °C. Overwintering occurs during the larval stage. A high mortality is observed at lower and higher temperatures, with the lowest mortality at about 17 °C [47,48]. Temperatures << 8 °C are not lethal, as even the icing of pipes does not eliminate the chironomids. This corresponds to the observed development of *P. grimmi* with mass development in August to November and indicates a high dependency of fertility with temperature. A calculation of the generation number in three months indicates an increase from a 0.7 generation at 12 °C, a 2.9 generation at 17 °C, and a 4.4 generation at 22 °C; thus, an exponential increase occurs, overlayed by mortality and egg number.

### 4.4. Diversity and Abundance

In DWDS, many different taxonomic groups of invertebrates are found worldwide, and several papers describe the species [49,50]. It is necessary to distinguish: (i) the origin of species (groundwater, filter-, and surface-water organisms), (ii) the size of the animals (macro-, meio-, and microinvertebrates), and (iii) the number (abundance) of pipe inhabitants (normal, increased, and mass development).

The introduction of small invertebrates into DWDS can occur by several routes: (i) introduction via a drinking-water treatment plant, mainly filter organisms; (ii) introduction via not completely closed water-storage tanks; (iii) the back-siphonage of surface water via unprotected connections, e.g., by a fire worker, the use of mobile standpipes, and the filling of the water tanks of ships; (iv) contamination during pipe construction and repair; and (v) the leakage of pipes when the pressure inside the pipe is temporarily lower than that outside.

The very high abundance of invertebrates in drinking-water networks presented in this paper is supported by several other studies [5,6,23,51–53].

For a few years, mass development of water lice (*Asellus aquaticus*) has been observed in several DWDS in northern Germany with abundances of >20 ind. $m^{-3}$ at hydrant monitoring, a density that has been classified as a guidance value for mass development [54]. Overall, 5% of the hydrant samples showed water lice densities of >100 ind. $m^{-3}$ (with a maximum of 869 water lice $m^{-3}$). Several authors report water lice densities of up to 1000 [5,23,52]. It must be considered that, with hydrant sampling, only parts of the water lice population are flushed out (about 15% [53]) because water lice climb to the pipe wall and move into the laminate flow layer.

The reasons for the recent mass development of water lice in DWDS are only partly known, and it seems to be a multi-factorial process, which includes brood care of the females, missing food limitation (water lice are omnivores) [7], and the occurrence of a third generation per year. Water lice grow at >4 °C, maturation starts at 7 °C, and at 12 °C, reproduction begins; thus, temperature effect for growth is significant. An increase in water temperature leads to faster growth of water lice and, as a consequence, earlier reproduction. The ancient two-generation cycle of water lice in North and Middle Europe described by several authors [55,56] has been changed by lake heating. The first report of the occurrence of three water lice generations in lakes per year was completed by Chambers [57].

Other macroinvertebrates have also been reported as developing a high abundance, e.g., oligochaetes with a median density of 200 and a maximum of 10,000 ind. $m^{-3}$ or chironomid larvae and snails (>1500 ind. $m^{-3}$) [5,7].

Meioinvertebrates such as Cladocera and copepods have been found in median concentrations between 600–750 ind. $m^{-3}$ with a maximum of 10,000 ind. $m^{-3}$; water mites reached a median of 150 and a maximum of 5000 ind. $m^{-3}$ [5].

Microinvertebrates are present in all drinking-water networks and reach very high densities of $5 \times 10^7$ to $7 \times 10^8$ [58,59].

*4.5. Health Risk*

The association of harmful bacteria with invertebrates in DWDS has been discussed since 1990 [60], and there are currently two aspects that have become the focus of further research: the invertebrates' gut biome as a vector of harmful microbes and invertebrates' excretion products and their role in triggering microbe development.

The occurrence of microbes in protozoa is well-known, mainly in some free-living amoebae (FLA), e.g., *Acanthamoeba castellanii,* and in ciliates, e.g., *Tetrahymena pyriformis* and *Cyclidium* spp. [19,61]. Inside *Acanthamoeba castellanii* as well as *Legionella pneumophila*, *E. coli* cells remain viable and can multiply [18,61]. Amoebae serve as the Trojan Horse of the microbial world [62], and if they occur in DWDS, the potential health risk is increased. Protozoa such as amoebae and ciliates feed on bacteria by phagocytosis, which means the bacteria are transferred into food vacuoles. Many bacteria can survive the ingestion process or are even able to multiply within the vacuoles. Three groups can be differentiated: those that survive within the amoebae without multiplying (some coliforms), those that multiply within the amoebae without causing cellular lysis (e.g., *Vibrio cholerae*), and those that multiply and cause cellular lysis in amoebae (e.g., *Legionella* spp. and *Listeria* spp. [19]). *Acanthamoeba* spp. serve as a vehicle for *Legionella pneumophila*, which means that *Acanthamoeba* feed on *L. pneumophila* cells by phagocytose and transport the cells within the digestive vacuoles [63]. The survival and transmission of *L. pneumophila* in DWDS is strongly linked to the presence of amoebae in water since FLA favor the multiplication of *L. pneumophila* occurring inside their digestive vacuoles [19].

FLA can colonize and regrow in drinking-water networks. In 26 studies from 18 different countries, eight DWDS were settled by FLA [64]; the data gave a frequency of only 3.4% in hydrant samples in the Northern European lowland, but this value can be underestimated because no specific FLA analysis was caried out. Many studies show moderate FLA densities of <5000 ind. $m^{-3}$; mass developments were registered in dead-end pipes in

summer with 815.000 ind. m$^{-3}$ [65]. The temperature increase will raise the health risk due to proliferation of FLA; the highest concentrations of *Acanthamoeba* were observed in the months from August to October [65].

Analyses of microbes' linkage to invertebrates' occurrence in DWDS were conducted, and it was pointed out that bacteria attached to the outer surface of the invertebrates do not represent a major source of bacterial contamination, but it was proven that they do inside the invertebrates at a mean of 4000 CFU per single invertebrate (among others, amphipods and copepods) [66]. Some invertebrates can serve as a potential vehicle for pathogenic microbes, and several reports point out that nematodes contain viable harmful bacteria. Levy et al. [67] reported 10–100 CFU microbes in a nematode, while Wolmarans et al. [68] found up to 4000 CFU per invertebrate. Additionally, egg masses of chironomids have been found to serve as a reservoir for harmful bacteria such as *Vibrio cholerae* and *Aeromonas* spp. [2,68], but no data about *E. coli* are available (personal communication). This phenomenon, in which bacteria enable continued existence in a hostile environment such as mucus of eggs, may not be restricted to only chironomids [69].

Water lice are also a potential carrier of *E. coli* and other harmful microbes; some bacteria such as *Pseudomonas* spec. and *Aeromonas hydrophila* live long term in the gut of *Asellus aquaticus* and were even found after 6 weeks of starvation [70], while *E. coli* were found only when they were present in the water with a small concentration of 350 *E. coli* in one water lice [71].

Water lice represent other risks for water quality in DWDS: (i) by accumulation of the very stable water lice feces (>6 weeks in a drinking-water stream) and of dead water lice in house filters, which lead to microbial growth exceeding drinking-water quality standards, especially with stagnant water conditions [7]; (ii) by promotion of potential human–pathogenic germs (*Pseudomonas aeruginosa*) through metabolic products of the water lice [24,72]; and (iii) by protection of microbes in the gut of *Asellus aquaticus* against disinfection measure [19,66,70,71].

## 5. Conclusions and Outlook

Climate change effects are mainly discussed for surface water and its impairment of the water quality. Long-term increases in DWDS are observed, and frequently temperatures >22 °C are already observed, without any extended temperature-monitoring programs; drinking-water temperature-limit values are given with 25 °C [41], respectively, more restrictive with 15–20 °C [42–44].

Increased insolation and an increased number of urban hot tropical days with temperatures >30 °C will occur, and it must be assumed that a corresponding increase of water temperature in DWDS will take place. Drinking-water pipes in urban areas are mainly constructed beside roads, which are often heavily heated by insolation, and temperature increases often reach down to water pipes. Thus, a further increase of invertebrate abundance must be expected, and unsealing of urban surfaces and promotion of urban green practices are one target in DWDS management.

Pipe inhabitants classified as macro-, meio-, and microfauna possess very high abundances; even the macroinvertebrates (>2 mm) reach 16 ind. m$^{-3}$ as median (90th percentile = 135 ind. m$^{-3}$) and lead at least to an aesthetic impairment of water quality. Abundances of meiofauna (0.1–2 mm) with 1700 ind. m$^{-3}$ as median and microfauna (0.025–0.1 mm) with 75,000 ind. m$^{-3}$ (median) point out clearly that biological drinking-water quality must become higher priority in DWDS management. In case of water lice, the very high population densities of up to 869 ind. m$^{-3}$ are reached due to a third generation per year as a consequence of increased temperatures in drinking-water pipes. We must expect that other organism such as chironomids (midges) also react in a simmer manner.

Advanced pipe-flushing technologies have become more important and are a necessary tool for drinking-water network management [73]. Studies on optimization of flushing parameters, such as sampling and quantifying invertebrates in DWDS [53,74,75], removal of soft deposits [76], flow velocity [77], or deposit formation [78], are available. Some

new flushing methods have been developed in the last few years, such as ice-pigging, an injection of a water–salt–ice mousse (Suez Water Technologies, Thame, UK), air scouring as an introduction of air into the water flow to create a high velocity and a turbulent flow, $CO_2$-flushing as a flushing with $CO_2$-enriched water to anesthetize the invertebrates so they cannot cling to the pipe wall [53], and umbrella flushing, which is the use of an umbrella as a cleaning pig [79].

It must be pointed out that measures against the impairment of water quality in DWDS by climate change cannot be implemented in the short term. Wells, water-treatment plants, and water-distribution networks have a very long service life; thus, technical adaptations to climatic impairment can only be made correspondingly slowly, e.g., by lowering drinking-water pipes to reduce temperature increases or by separating complex DWDS into smaller sub-units that enable more targeted pipe flushing.

The knowledge of invertebrates' settlement in DWDS is of high significance for any network management and is the basis to securing drinking-water quality. However, overall monitoring of invertebrates requires standard methods, which allow filtering of even fine organisms without hard skeletal structures, such as nematodes and amoebas, at a high flow rate of at least $0.5 \text{ m s}^{-1}$ or better yet at $1 \text{ m s}^{-1}$ [25]. Adapted filters have been developed as stainless steel, low-pressure, high flow-through filters [26], and DWDS flushing strategies must be linked with network modeling, e.g., for daily flow dynamics, sedimentation, and accumulation of organic matter [78].

New analytical methods are based on DNA and RNA extraction and sequencing as well as by using some primers for taxonomic groups. These methods allow a sensitive detection of species such as animal groups and are very useful in the case of harmful invertebrates, but for a quantitative analysis of species, they are not yet a standard method [18].

Today, there is a broad consensus that the goal of good drinking-water quality is not fulfilled at the point of transfer from the treatment plant to the DWDS but only at the point of connection to the consumer [3,24]. Thus, water suppliers need to focus more on DWDS management, which comprises temperature registration within DWDS, microbial and invertebrate monitoring, water flow modeling, and pipe flushing with control by turbidity and invertebrate discharge [80].

Further research must analyze the key factors of the development of invertebrates in drinking-water networks, the effects of increased water temperatures, and the interactions of invertebrates and microbes. The available data clearly highlight the impairment of water quality in DWDS due to increasing temperatures.

**Author Contributions:** All authors contributed to the work presented in this paper and participated on project management. G.G. evaluated the data of field research, U.M. directed the biological laboratory analyses, and M.S. provided the drinking water network flushings. As specific contribution G.G. designed the article content and wrote the manuscript. All authors have read and agreed to the published version of the manuscript.

**Funding:** This research received no external funding.

**Acknowledgments:** This work was made possible by the support of various drinking water suppliers who must remain anonymous due to the sensitive drinking water data. Thanks are due for the provision of resources and openness to systematic investigations in the drinking water networks.

**Conflicts of Interest:** The authors declare no conflict of interest.

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
