# Peer review of "Climate Change: Water Temperature and Invertebrate Propagation in Drinking-Water Distribution Systems, Effects, and Risk Assessment"

_water, doi:10.3390/w14081246_

Round 1
Reviewer 1 Report
The comments and suggestions for the authors are in the uploaded file.

Author Response
Introduction
Line 39: ‘anesthetic …’ Incorrect English: unaesthetic. changed
Line 44-83: overview of climate change and temperature impact on biological processes: connection with invertebrate propagation is not clear or disputable. This is given only for bank filtration, inserted L 68
Line 84-88: I miss a realistic climate risk: floods and water intrusion. Inserted L73-75
Line 89: change of water quality due to biological activity is biological Instability. This aspect was not presented by the authors, in fact the authors hardly reflect on water quality effects. Specified, L 104
Line 102-103: ‘development is regulated by water temperature…..’ I miss here the role of the nutritional conditions. This in inserted, L122 – 127.
Line 96-123: key factors are the growth factors (nutrients and other abiotic factors, including temperature) for the actors: bacterial and invertebrate communities. This text presents different aspects without a clear focus on the rational and objectives of the current study. A revision of this part is required. It reads too much as a literature review in the discussion but with no clear focus. Furthermore I miss a reflection on an important influence of chlorine on the processes in DWDS. This part is revised L 122 - 134
Methods
Line 131: reservoirs as source of raw water……..: unclear – changed L 153. Surface water from ……….changed L 154
Line 131-132: chlorination has not been performed: what is meant here? In the DWDS or in the treatment??? Are all sampled DWDS non=chlorinated? Changed L 153
Line 133: conspicuous: specification required. Unclear statement. Changed L 156
Line 137: ‘all pipe types’ Specification required. No significant effect: reference required and statistical test results. I have changed the description of studies pipe parameters, L 160-168
Lines 136-146: all relevant factors affecting invertebrate populations not included in the study. Large omission which will have an impact on the conclusions of the current study. See above
Line 154: Final filtrate volume: is that the final volume of the filtered invertebrate suspension? Unclear statement here. ‘restricted to less than …’ Incorrect English: the filtered loose deposits including invertebrates are resuspended in a sample volume of 300 mL and preserved with ethanol to a final ….changed
Line 161-162: ‘spreading and developed into water lice’ Incorrect English sentence. changed
Line 187-188: No testing for the normal or lognormal distribution of the data: Pearson versus spearman rho. Please explain why there was no testing on normality. Invertebrate distribution is characterized by hot spots due to local conditions (pipe corrosion, dead end pipes, slide valve domes) and not be a normal distribution.
Results
Line 190-225: confusing text. Title is about drinking water reservoirs but in the first sentence the author speak of ‘surface water reservoirs’: specificity required. Sorry, but I have not used surface water in the text Furthermore, after reading the methods the reader expects to get information on invertebrates in drinking water distribution systems. Instead the results start with a literature review and no study data about the effect of water temperature on surface water reservoir biology, cyanobacteria and taste and odor, and a invertebrate problem in a drinking water reservoir while these were not mentioned in the methods! I deleted the information about cyanobacteria in the Results and Discussion, now I only name it with references. I changed the title including water temperature, and the water temperatures of the DWDS are own experimental data.
Line 227: English, long-time should be long-term. changed
Line 240: English: short-time should be short-term! But I would like to call this seasonal temperature variation rather than short-term variation! Changed. Why not start with this seasonal variation and describe the long-term effect? Ok, done. Furthermore the authors assume that the rise of both raw water and soil temperature are responsible for the water temperature increase in Fig. 1 but they do not give data which could confirm that both effects are responsible. Now given by 3.2.1. and 3.2.2.
Line 251: daily course is not proper English! A daily course refers to education???? changed
Next: ok, now the authors come with data on the effect of soil temperature (I still expect data on invertebrates??)
Line 262-264 changed and 273-275:this is supported by transport and distribution pipes speculations. About fig. 5: where are the residence time calculations? No residence times are given
Paragraph 3.3. Now the authors come to the results ………..Temperature data are own results, too, now named in the title
Line 303-305: statement without references. Furthermore, is this based on surface water biology or is this specifically observed in DWDS. Inserted surface water Be precise with statements: sources and connection with your data. In the following text the authors describe this connection: revision required. First describe the results of both invasive species and then your statement/conclusion! changed
Line 306: ‘refilling ship tanks’: we were talking about DWDS??? The authors confuse results with discussion. During refilling ship tanks, back siphoning can introduce new species from the ship tank into the DWDS. Hot spots of invasive species we found in the habour. So the title of this chapter should be Results and Discussion!
Line 310: ‘settled’ for Asellus aquaticus: should be populated! changed
Line 320-323: speculations: high abundance of invertebrates! What is low – moderate or high density of invertebrates in DWDS? Do the authors have references for this statement? Reference is given, The key factors presented are not part of this research so please give references for these key factors. Key factors are deleted
Line 326-327: ‘temperature is the only …’ I do not agree: there is also a seasonal variation in other abiotic and biotic factors!!! Revision required. The authors simplify the correlation with temperature to a one-dimensional correlation but that is NOT correct. There is also something called as co-correlation and the temperature impact is much more complex than assumed here. Quality of treated water (outflow of water works) is very constant, with exception of temperature. Ground water does not undergo seasonal variation, bank filtration water is constant, too, due to mixing of infiltration water and ground water, and high mountain reservoirs are old oligotrophic lakes without extrem seasonal variation. Direct abstraction of surface water is not done in the studies area.
Line 346: incorrect, long-time ….changed
Line 354: ‘but local yearly mean’ English: ‘but locally the annual mean ..’ changed
Line 362: Figure 10 should be Figure 9. ok
Line 363: (equaling the winter form) ??? Please explain: not clear. changed, = winter form
Line 364: 40 embryos per female: reference required. Own data, added with a semicolon
Figure 9: where is F0?? No explanation of the colored bars???? Sorry, mistake in the figure, changed
The authors assume a correlation of population growth of water lice with air temperature (no water temperature data available): this is speculation and there is no evidence presented that this growth is due temperature increase. The growth could also be caused by natural population of this network due to adaptation, positive environmental conditions (enough organic matter in combination with absence of net cleaning). In 2018-2019 the population density has reached its steady state. All over the years we found the same conditions during hydrant monitoring, no changes in water treatment etc. We describe the data, and only assume a correlation with temperature change. The assumption of a steady state after 21 generations is a very long period, too.
Discussion
Lines 379-432: this part is not dedicated to the title of the paper: could not reflect on the significant of all presented issues for the objective(s) of the overall paper since that was not presented in the introduction. But talking about the effect of climate change and temperature increase on water lakes and trying to connect this with the title of the paper is not appropriate and highly speculative. changed, title and abstract
Lines 454-439: repeated criticism on temperature as the key factor: see comment on lines 326-327. Think of co-correlation of temperature with other key factors such as food availability. The authors are too focused on temperature. Yes, and we observed no food limitation.
The following part describes temperature effects on diversity of invertebrates. That is ok in line with the title of the paper.
Line 506-516: here again issues which are not relevant for invertebrates in DWDS. deleted
Line 517-568: relevant information with respect to the association of invertebrates with pathogens in drinking water especially when there is no DWDS disinfectant dosed. For Legionella the relevance of this association this has been proven convincingly. For the other associations and pathogens the relevancy is still not clear at the moment.

Reviewer 2 Report
Dear Authors
The current manuscript (ID: water-1634270) under the title “Climate change: Invertebrate Propagation in Drinking Water Distribution Systems, Effects, and Risk Assessment”, paper concept is very attractive and provides a summary of the knowledge and challenges concerning the development of different invertebrates in DWDS. While the presented paper needs some Minor modification.
- In the abstract: Lines 6-7: Change “DWDS). Macro- and meio invertebrates” to "Macro-, meio-, and Micro-invertebrates"
- At the end of the Introduction part, authors must clearly provide the “ aims of the current work”
- In the Results part (3.1. Temperature increase in drinking water reservoirs) authors tacking about cyanobacteria (Lines 91-93). Is this given data related to the current paper? If yes, the authors must provide the methodology of this part as a primary-assistant-tool. On another point, the authors mentioned some refs in this part. I recommend moving these References to the Discussion Part.
- In Fig. 2, to improve data presenting, it will be better if authors use any modeling tool
- Authors must add a Conclusion Part
- Authors must follow the Water guideline.
- In the line of the impact of climate changes on the planktonic organisms, authors may use the following refs:
- - https://doi.org/10.3390/jmse9121328
- - https://doi.org/10.3390/d13060268
All the best.
The Reviewer #
Author Response
- done
- changed, L 137-140
- I deleted the cyanotoxin data, in 3.1 Results and discussion, because I do not mention cyanobacteria in the Discussion, I still give some references in 3.1
- in Fig. 2 (now fig. 4), I calculated the regression line and give the in increase, overall it is only a trend analysis and no significance analysis
- Conclusion is added
- Water guidelines are applicated
- The references due to climate change effect on plancton oargniasms are related wih marine plancton, we cannot apply these data to limnic organismsn.
Reviewer 3 Report
The manuscript concerns an impoertant summary of the knowledge and challenges concerning development of invertebrates in drinking water distribution systems. In nearly all DWDS macro- and meioinvertebrates were found. Data about diversity and abundance point out clearly a high probability of mass development and invertebrate monitoring must be focus of any DWDS management. DWDS water temperature is increasing by climate change effects, and as a consequence growth and reproduction of invertebrates is increasing. Seasonal development of a chironomid (Paratanytarus grimmii) and longtime development of water lice (Asellus aquaticus) are given. Remarks: line 4: lack of affiliation and name of the third author: Michael Scheideler. Better quality of the following figure should be provide: Figure 3. Seasonal heating of drinking water in Germany/Lower Saxony (DWDS B); water works outlet temperature, the temperature of the soil surface and the temperature increase of the drinking water pipe are all given; pipe DN in 1.15 m depth. Line: 250, double dot at the end of the sentence. What is the minimal depth of the pipe (give information of some regulations)? Line 786: Lack of publication title: ref. no. 81. Consider adding the English title to the publications in the reference, eg. ref. nos. 78, 80, etc.
Define the type of the paper: review.
Add reference to the figures in in case you are not the Author of it.
The review's significant achievements and recommendations should be highlighted in the conclusion.
Author Response
Remarks: line 4: lack of affiliation and name of the third author: Michael Scheideler. OK, done
Better quality of the following figure should be provide: Figure 3. Seasonal heating of drinking water in Germany/Lower Saxony (DWDS B); water works outlet temperature, the temperature of the soil surface and the temperature increase of the drinking water pipe are all given; pipe DN in 1.15 m depth. I changed the figure 3 to higher quality.
Line: 250, double dot at the end of the sentence. Deleted. What is the minimal depth of the pipe (give information of some regulations)? The pipe depth is constant for this section, in the net, some pipes are situated in other depths, but this is not relevant for the monitoring station. Line 786: Lack of publication title: ref. no. 81. Changed, now 79. Inkinen ...Consider adding the English title to the publications in the reference, eg. ref. nos. 78, 80, etc. Done
Define the type of the paper: review. The other reviewer donot advise this as a review paper, and without cyanobactria my own work is more in focus.
Add reference to the figures in in case you are not the Author of it. I am the author of the figures.
The review's significant achievements and recommendations should be highlighted in the conclusion. Done
Round 2
Reviewer 3 Report
Not all of my previous remarks have been addressed:
- The review's significant achievements and recommendations should be highlighted in the conclusion. "Done"
What is the achievement of the presented manuscript in comparison to the previous publications: [13] and [37]?
Author Response
Dear reviewer, I inserted Line 619 - 627 in the Conclusion to highlight the results. And renamed it to Conclusion and Outlook.
Overall I revised the paper, e.g. bei Table 1 - water quality data, and Fig. 9, population increase to show that no steady state is reached.
Paper 13 - Gross-Wittke et al. , presents enclosure experiment in Lake Tegel, Berlin for experimental studies of water heating in bank filtration. In this paper the experimental results are presented.
Paper 37 - Gunkel et al., gwf is a German journal for technicans and stakeholders, no scientific journal, an advertisement based journal, this paper focus on the significance of climate change effects as challenges in drinking water networks, without detailled information of our experimental work.
Track changes in the 2nd revision
L 74-75, inserted: direct water abstraction….
L 124, deleted
L 134, inserted: this means….
L 139-141, inserted: Overall….
L 203-204, inserted: DWDS E ….
L 205: Table 1 new
L 198 Reference (30) for size classification
L 246: inserted: without micro-sieving
L 247: inserted: adults are…..
L 259: more precise described
L 317-321: inserted significance of water residence times
L 360-368: new order
L 369 ff: inserted, The very high….
L 421-423: inserted: spatial extension…
L 433, inserted fig. 9
L 440-445: description of no steady state
L 464-465 inserted: This estimation
L 685-693: inserted conclusion highlights
Concerning English language I used the MDPI language revision system, andI think with your remarks it is now ok.